# Stability, Structure, Rheological Properties, and Tribology of Flaxseed Gum Filled with Rice Bran Oil Bodies

**DOI:** 10.3390/foods11193110

**Published:** 2022-10-06

**Authors:** Xiaoyu Li, Qiuyu Wang, Jia Hao, Duoxia Xu

**Affiliations:** 1Beijing Advanced Innovation Center for Food Nutrition and Human Health, Beijing Technology and Business University, Beijing 100048, China; 2School of Food and Health, Beijing Technology and Business University, Beijing 100048, China; 3Beijing Engineering and Technology Research Center of Food Additives, Beijing Technology and Business University, Beijing 100048, China; 4Beijing Higher Institution Engineering Research Center of Food Additives and Ingredients, Beijing Technology and Business University, Beijing 100048, China; 5Beijing Key Laboratory of Flavor Chemistry, Beijing Technology and Business University, Beijing 100048, China; 6Beijing Laboratory for Food Quality and Safety, Beijing Technology and Business University, Beijing 100048, China

**Keywords:** rice bran oil body, emulsion-filled gel, physical characterization, structure, rheological properties, tribology

## Abstract

In this study, rice bran oil bodies (RBOBs) were filled with varying concentrations of flaxseed gum (FG) to construct an RBOB-FG emulsion-filled gel system. The particle size distribution, zeta potential, physical stability, and microstructure were measured and observed. The molecular interaction of RBOBs and FG was studied by Fourier transform infrared spectroscopy (FTIR). In addition, the rheological and the tribology properties of the RBOB-FG emulsion-filled gels were evaluated. We found that the dispersibility and stability of the RBOB droplets was improved by FG hydrogel, and the electrostatic repulsion of the system was enhanced. FTIR analysis indicated that the hydrogen bonds and intermolecular forces were the major driving forces in the formation of RBOB-FG emulsion-filled gel. An emulsion-filled gel-like structure was formed, which further improved the rheological properties, with increased firmness, storage modulus values, and viscoelasticity, forming thermally stable networks. In the tribological test, with increased FG concentration, the friction coefficient (*μ*) decreased. The elasticity of RBOB-FG emulsion-filled gels and the ball-bearing effect led to a minimum boundary friction coefficient (*μ*). These results might contribute to the development of oil-body-based functional ingredients for applications in plant-based foods as fat replacements and delivery systems.

## 1. Introduction

Oil bodies (OBs), also known as oleosomes, are micron- or submicron-sized natural oil droplets [1]. OBs are mainly composed of triglycerides (TAG) covered by phospholipids and proteins, including oleosins (15–26 kDa), caleosins (25–35 kDa), and steroleosins (40–55 kDa) [2,3]. The hydrophobic domains of these endogenous and exogenous proteins are embedded inside the TAG core, and their hydrophilic ends face towards the aqueous phase in the cytoplasm. The phospholipid–protein layer can improve the physicochemical stability of OBs [4]. At present, the extraction methods of oil bodies mainly include aqueous extraction and enzyme-assisted extraction. The aqueous extraction of OBs requires seeds to be soaked in an aqueous medium, followed by blending or pressing to disrupt cell walls and release intracellular materials. Urea, sucrose, deionized water, salt, alkali, and buffer solution (including Tris–HCl and PBS) are often used as the grinding medium for aqueous extraction [5]. For enzyme assisted extraction, because plant cell walls are composed of cellulose, hemicelluloses, lignin, and pectin, cellulase, hemicellulose, pectinase, xylanase, and β-glucanase could also destroy the cell wall. However, the high specificity of enzymes considerably limits the degree of hydrolysis, and OB-associated proteins may be destroyed into small peptides through hydrolysis [6].

Rice bran oil bodies (RBOBs) are natural pre-emulsified emulsions composed of a triacylglycerol core surrounded by a phosphate layer embedded with various kinds of oleosins and some minor proteins of higher molecular mass [7]. Extracted RBOBs contain most of the bioactive substances in rice bran. Among them, the contents of tocotrienol, tocopherol, and oryzanol were reported to be 77%, 73%, and 91%, respectively [8]. Few studies on RBOBs have been conducted to date, but the processing and reuse of rice bran as a byproduct has a considerable influence on oil production [9]. Due to their nutritional value and natural emulsifying properties, OBs can replace oil or emulsion droplets [10]. Flaxseed gum (FG) is composed of 75% polysaccharides and 25% acid polysaccharides [11]. Owing to its functional properties, FG has been widely used for thickening and swelling and as an emulsifier [12]. However, reports on changes in gel properties and mechanisms of FG gel matrices induced by RBOBs used as filler are still extremely limited.

Emulsion gel contains both emulsion droplets and gels [13]. Emulsion-filled gel is a kind of emulsion gel, with oil droplets in the gel matrix as filler particles [14]. According to their effect on gel properties, droplets can be classified as either active or inactive fillers. Active filters are bounded to the gel network and contribute to gel strength, whereas inactive fillers are difficult to bind with the gel matrix and do not effect gel viscoelasticity [15,16]. The gel properties of emulsion gels are strongly influenced by the structure, molecular weight, and concentration of the polymer, as well as the interaction between the gel matrix and the number of padding droplets, droplet size, and stiffness [17]. Owing to its multifunctional structure and composition, emulsion gel has considerable potential for applications in the food industry. Several applications have been proposed for emulsion-filled gels as fat/saturated fat replacements.

In this study, RBOBs in rice bran were extracted by the aqueous method. Two types of RBOB-FG emulsion-filled gels system (fluid emulsion gels/bulk emulsion gels) were constructed using three concentrations of FG (0.2, 0.8, and 2.0 wt.%). The particle size distribution, zeta potential, physical stability, and microstructure of the RBOB-FG emulsion-filled gels were measured and observed. The molecular interaction of RBOBs was studied by Fourier infrared spectroscopy. The rheological properties and the tribology of the RBOB-FG emulsion-filled gels were also evaluated. This research may contribute to the development of a new and simple strategy to determine the structure of RBOBs, which can be used in the development of fat replacements and edible soft solid materials.

## 2. Materials and Methods

### 2.1. Materials

Rice bran was purchased from Shuyang Runyi Agricultural Technology Co., Ltd. (Shuyang, China). FG was obtained from Yuanye Biological Company (Shanghai, China) and used without further treatment. The FG contained 70.92% polysaccharide, 12.67% protein, and 16.31% ash, as supplied by the manufacture. Nile blue A and Nile red were provided by Sigma-Aldrich Co. (St. Louis, MO, USA). All other chemicals were of analytical grade. All solutions and emulsions were prepared using ultrapure water (SMART-N, HealForce, Shanghai, China).

### 2.2. Extraction of Rice Bran Oil Bodies

Coarse OBs were physically isolated from a homogenate of rice bran according to the method described in [18], with a slight modification. Briefly, unexpanded rice bran was passed through a 20 mesh screen to obtain a more uniform raw material. The rice bran was soaked in 10.0 mmol/L phosphate buffer solution at pH 7.0 with a solid–liquid ratio of 1:5 and stored overnight at 4 °C. The raw material was stirred and sheared at 10,000 rpm high speed for 3 min by a high-speed mixer (ULTRA TURRAX T25 digital, IKA, Staufen, Germany) and magnetically stirred in a water bath at 50 °C for 3 h. Then, it was centrifuged by centrifuge (Avanti JXN-26, Beckman, Indianapolis, USA) at 10,000 rpm at 4 °C for 20 min. A spoon was used to obtain the upper layer of the centrifuged samples containing the RBOB cream.

### 2.3. Preparation of RBOB-FG Emulsion-Filled Gel and FG Hydrogel

An RBOB emulsion was prepared by dispersing extracted oil body creams into phosphate-buffered solution (10 mM, pH 7.0) under magnetic stirring. Varying contents of FG were mixed into RBOB emulsion, heated at 50 °C in a water bath, and magnetically stirred for 1.5 h. The final oil concentration was 5.0 wt.%, and the concentration of FG was 0.2, 0.8, and 2.0 wt.% for each of the mixtures, respectively. The pH of the RBOB-FG emulsion-filled gels was determined to be 6.5. Then, all samples were stored in refrigerator (4 °C, 12 h) for gelation to obtain RBOB-FG emulsion-filled gel. For comparison, FG hydrogel without RBOBs was prepared as a control at pH 6.5, using NaOH (1.0 M) and HCl (1.0 M). Concentrations of 0.2, 0.8, and 2.0% FG were used for the gel matrix because we expected to obtain liquid and solid-like gel samples for food applications.

### 2.4. Particle Size and ζ-Potential Measurements

For analysis of particle size and ζ-potential, average particle size and distribution were measured using SDC-Microtrac S3500 laser diffraction equipment (Microtrac, Montgomery Ville, PA, USA) with 3 readings for each sample. Zeta potentials of the droplets were measured with a Nano-ZS90 zetasizer (Malvern Instruments, Worcestershire, UK) through the assessment of droplet velocity and the direction in the electrical field. Phosphate-buffered solution (10.0 mmol/L) was used to dilute the measured samples with the same pH as the samples to avoid multiple scattering effects. All samples were equilibrated for 120 s before obtaining data.

### 2.5. Physical Stability Measurements

A LUMiSizer (LUM GmbH, Berlin, Germany) was used to measure the physical stabilities of RBOB-FG emulsion-filled gel, RBOB emulsion, and FG hydrogel. After storage at 4 °C storage for 12 h, samples were accelerated to test their instability (25 °C; 0.4 mL; 4000 rpm; 4.25 h; 60 s time interval) according to the method described in [19].

### 2.6. Fourier Transform Infrared (FTIR) Spectroscopy

A Bruker Tensor II instrument (Waltham, MA, USA) was used to acquire infrared spectra of RBOB-FG emulsion-filled gel, FG hydrogel, and RBOBs at 25 °C. PBS was used as the blank. All samples were freeze-dried with KBr and pressed into thin sheets. Samples were tested in the scanning range of 500–4000 cm^−1^ in absorption mode, with 32 scans and a resolution of 4 cm^−1^.

### 2.7. Microstructural Analysis

#### 2.7.1. Scanning Electron Microscopy (SEM)

The microstructure of dried and structured RBOB-FG emulsion-filled gel samples were analyzed using scanning electron microscopy (S4800, Hitachi, Tokyo, Japan). The accelerating voltage was 1.2 kV during the measurements. A small portion of sample was placed on an aluminum holder using a double-adhesive conductive pad. Loose particles were removed by spraying dry air on the stent surface. RBOBs, RBOB-FG emulsion-filled gel, and FG hydrogel were observed with an IX81 light microscope (Olympus, Tokyo, Japan).

#### 2.7.2. Confocal Laser Scanning Microscopy (CLSM)

A CLSM (FV3000, Olympus, Tokyo, Japan) was used to observe the microstructure of samples. The TAG cores of RBOBs were stained with Nile red, and RBOB proteins were stained with Nile blue A [20]. All samples were observed using a 10× eyepiece under 60× objective lenses (oil immersion). Digital image files were obtained in 1024 × 1024 pixel resolution.

### 2.8. Microrheological Behavior

The microrheological properties of samples were measured by Rheolaser Master (Formulation, l’Union, France) based on the diffusing wave spectroscopy theory. A 3 h microrheology test was carried out on RBOBs, RBOB-FG emulsion-filled gel, and FG hydrogel.

### 2.9. Rheological Properties

After the gelation (4 °C, 12 h), the rheological properties of the FG hydrogel/RBOB-FG emulsion-filled gels were characterized at 25 °C on a Haake 6000 RheoStress rheometer (Thermo Scientific, Waltham, MA, USA). A 35 mm diameter parallel plate with waterproof sandpaper was used for measurement [21]. The viscoelastic properties of the RBOB-FG emulsion-filled gel/hydrogels were determined using a 1.0% fixed strain and frequency scanning in the range of 0.1 to 100 rad/s. The apparent shear viscosity of the RBOB-FG emulsion-filled gel/hydrogels was measured at shear rates ranging from 0.1 to 100 s^−1^. The experimental curve was fitted using the Herschel–Bulkley model represented by $\tau =\tau_{0}+k\gamma^{n}$, where τ is the shear stress (Pa), $\tau_{0}$ is the yield stress (Pa), γ is the shear rate (s^−1^), K is the consistency index (Pa s^n^), and *n* is the flow behavior index.

Before starting the experiment, low-viscosity silicone oil was used to prevent the water from evaporating. Samples were loaded onto the rheometer, which was preheated to 90 °C. After an equilibration time of 5 min, a temperature sweep was processed. A cooling step from 90 to 20 °C and a heating step from 20 °C to 90 °C were performed at a constant rate of 1 °C/min. The process was examined by monitoring the G′ (storage modulus) and G″ (loss modulus) change under a fixed frequency (1 Hz) and 1.0% strain.

### 2.10. Tribology Measurement

A TA-DHR friction rheometer was used with a full-ring stainless steel probe, and the surface of the oral cavity was simulated with polydimethylsiloxane (PDMS) to measure the lubricity of the particles. The rotating sphere automatically adjusted with a normal force (3 N) evenly distributed on the lower plate at 37 °C. As the sliding speed increased from 0.1 to 450 mm/s, the friction force between the stainless ball and the plates was recorded. The coefficient of friction was derived using the following formula: coefficient of friction = friction force/set normal force.

PDMS (Sylgard184) production method: mix base fluid and cross-linking agent (10:1 *w*/*w*) to prepare PDMS (Sylgard184) with a surface roughness (Ra) < 50 nm, vacuum to remove bubbles generated during mixing, and smooth container stainless steel mold; cure overnight at 70 °C. A rotational rheometer mounted with an accessory based on 3-ball plate tribology geometry was used for tribological tests [22].

### 2.11. Statistical Analysis

The rheology and tribology measurements were repeated twice, and other measurements were repeated in triplicate. The plots were drawn using Origin 8.5 software (OriginLab Co., Northampton, MA, USA).

The average values and the standard deviation (SD) were reported using descriptive statistical analysis, which was performed through SPSS statistics, version 25 (IBM Inc., Armonk, NY, USA).

## 3. Results and Discussion

### 3.1. Effect of FG Concentrations on the Particle Size and Zeta Potential of RBOBs

The particle size distributions of RBOB-FG emulsion-filled gel formed by RBOB emulsions mixed with varying FG concentrations (0.2, 0.8, and 2.0 wt.%) at pH 6.5 are shown in Figure 1a. At pH 6.5, the particle size of the RBOB emulsions without FG was around 10 μm, which is much larger than the RBOBs in the FG hydrogel matrix, indicating that the RBOBs were aggregated, possibly because the ion dipole changed leads to the interactions to overcome repulsions [23]. Electrostatic repulsion between RBOBs is insufficient to overcome various attractive interactions, e.g., van der Waals forces and hydrophobic interactions between OB proteins [24]. However, after FG was added to the RBOB emulsion, the droplet size of the RBOB emulsion decreased to around 2–6 μm. A non-uniform particle size distribution was observed when 2.0% FG was added to the RBOB emulsion, possibly due to the excessive FG concentration as a continuous phase, leading to high viscosity and steric hindrance effect in the emulsion droplet-filled gel system.

The ζ-potentials of RBOB-FG emulsion-filled gels are displayed in Figure 1b. As shown, the ζ-potential decreased from about −27 mV to −47 mV for RBOB-FG emulsion-filled gel with increased FG concentration. After the FG concentration reached 0.8 wt.%, the ζ-potential did not differ significantly, indicating that about 0.8 wt.% FG-stabilized RBOBs droplets had sufficiently negative charges, increasing the electrostatic repulsion between droplets, as proven by physical stability (Figure 1c). After all samples were processed by a LUMiSizer, obvious stratification and sedimentation were observed in RBOB emulsions. Profiles lay at the bottom, indicated as red, and the final profiles lay at the top, indicated as green [25]. Emulsion particles move individually at different speeds. The profiles were closely spaced with considerably shorter distances. A diffuse sedimentation front moves with considerably slower velocity, indicating a swarm or polydisperse sedimentation occurrences. When the FG concentration decreased to 0.2 wt.%, the droplets movement of RBOBs was suppressed. As with increased FG concentration, no movement of the droplets was observed during centrifugation. Therefore, the dispersibility and stability of RBOB droplets could be improved by appropriate concentration of FG hydrogel, enhancing the electrostatic repulsion between RBOB droplets. Additionally, FG as gel matrix provided a sufficiently high electrostatic repulsion and steric repulsion between the RBOB droplets, inhibiting aggregation [26]. Varying concentrations of polysaccharides in emulsion-filled gel exhibited varying textural behaviors. Fluid RBOB-FG emulsion-filled gel can be formed by adding 0.2 wt.% FG after homogenization and shearing, with the potential to form a microgel. Solid-like emulsion-filled gel can be formed by adding a concentration of 0.8 wt.% FG or higher to develop low-fat salad dressings or meat fat substitutes.

### 3.2. FTIR Analysis

The molecular interaction between RBOB emulsion and FG was studied by FTIR (Figure 2). FG exhibited an absorption peak characteristic of polysaccharides, with the FTIR spectrum indicating O-H stretching at 3600–3200 cm^−1^ and C-H stretching at 3200–2800 cm^−1^, as well as peaks characteristic of polysaccharides at 1644–1630 cm^−1^ and 1036 cm^−1^. The characteristic absorption peaks near 849 and 831 cm^−1^ in the spectrum indicated that there were glycosidic bonds [27]. With respect to the RBOB emulsion, the most intense peaks were located at 3050–2800 cm^−1^ and 1738 cm^−1^ (C-H and C=O stretch vibrations, respectively), 1454 cm^−1^(C-H deformation), and 1200–1100 cm^−1^(C–O bonding) [28]. RBOB-FG emulsion-filled gels showed several bands attributed to the presence of the RBOB emulsion, with the most intense peaks located at 2918 cm^−1^, 2851 cm^−1^, and 1744 cm^−1^. As portrayed in Figure 2, the successful blending of oil droplets and the FG matrix of the RBOB-FG emulsion-filled gels was confirmed by the characteristic peak at 1744 cm^−1^(C=O). A new peak appeared at 1711 cm^−1^, with the amide I and II bands of RBOB proteins shifted from 1644.07 to 1650.56 cm^−1^ and from 1548.34 to 1544.53 cm^−1^, possibly due to the hydrogen bonding interaction between RBOB emulsion and FG [29]. The above results demonstrate that FG and RBOB emulsion showed different interactions. Hydrogen bonds were the major driving forces in the formation of RBOB-FG emulsion-filled gel, improving the gel properties.

### 3.3. SEM Microstructure

The microstructures of RBOB emulsion, FG hydrogel, and RBOB-FG emulsion-filled gel microparticles were investigated by SEM, as shown in Figure 3. The freeze-dried SEM images indicate that the RBOBs were dispersed and tightly packed in droplet forms, reflecting a highly concentrated emulsion structure (Figure 3a). As the concentration of FG increased, the gel network formed by FG became denser (Figure 3b,d,f) due to the increased viscosity of FG in the solution [30]. After hydration, swelling, and rearrangement, an increased chance of collision and entanglement was observed between the FG molecules [31]. When the FG hydrogel was filled with RBOB emulsion, the RBOB droplets were tightly bound to the entangled chains and network nodes of FG (Figure 3c,e,g). The addition of RBOBs also resulted in an emulsion-filled gel structure as a result of the gelation of the water continuous phase containing FG, with RBOB droplets physically entrapped in the FG hydrogel matrix.

### 3.4. CLSM Microstructure

CLSM micrographs of RBOB-FG emulsion-filled gels/FG hydrogels with varying FG concentrations and neutral pH are shown in Figure 4. The RBOB emulsions were extracted under natural conditions [26]. After aqueous phase extraction, obvious aggregation was observed. The weak electrostatic repulsion and bridging effect between endogenous and exogenous RBOBs proteins led to considerable aggregation as an instable RBOB emulsion system. Generally, the RBOBs dispersed evenly in the hydrogel matrix, and a suitable concentration FG, as an anionic polysaccharide, could provide a sufficient electrostatic repulsive interaction and steric hindrance effect to disperse oil droplets; similar results were obtained with respect to particle size. Compared to samples with relatively low FG concentrations, small portions of oil droplet reaggregates were observed at 2.0 wt.% FG, which may be attributed to the fact that the increase in FG concentration in the continuous phase caused the gel network containing oil droplets to form a three-dimensional structure so that the oil droplets existed in different network layers; furthermore, RBOB-depleted flocculation may have been cause by additional unabsorbed FG molecules.

### 3.5. Microrheological Analysis

The microrheological properties of RBOB-FG emulsion-filled gels/FG hydrogels with varying FG concentrations were investigated (Figure 5). In the Rheolaser test, the Brownian motion was monitored, and no damage was caused to the structure of emulsion samples [32]. The mean square displacement (MSD) curves of RBOB emulsion droplets (Figure 5a) was nonlinear, indicating that the emulsion was viscoelastic, exhibiting liquid-like behavior. Furthermore, the MSD curves of RBOB-FG emulsion-filled gel (Figure 5c,e,g) and FG hydrogel (Figure 5b,d,f) were nonlinear, with platform areas appearing over time and MSD curves gradually widening with time due to the migration of particles, indicating viscoelastic behavior. However, more viscous behavior was observed in the MSD of RBOB-FG emulsion-filled gel, indicating that the movements of RBOB emulsion droplets in emulsion-filled gel were restricted. Owing to the interaction between polysaccharides and RBOB droplets, the range of particle movements of RBOB-FG emulsion-filled gels was reduced during the decorrelation time (t_dec_). The elasticity of the sample with shorter decorrelation time (0 < t_dec_ < 1.5 s) was represented by the height of the MSD curves. With a short decorrelation time, a downward movement of RBOB-FG emulsion-filled gel and FG hydrogel MSD curves was observed, indicating that collision between particles, viscoelasticity, and interactions in the gel network were gradually enhanced in the system.

The MSD curves show that a stronger elastic system of RBOB-FG emulsion-filled gel was formed. Compared to FG hydrogels, the droplet movements were limited within RBOB-FG emulsion-filled gel due to the high viscosity.

### 3.6. Rheological Properties

To study the effect of RBOB emulsion on the rheological properties of varying FG concentrations, a frequency sweep and temperature sweep were performed on the RBOB-FG emulsion-filled gels and FG hydrogels. Storage modulus (G′) and loss modulus (G″) are commonly used indicators to characterize the gel properties [33]. G′ refers to the elastic characteristics of a gel system, and G″ reflects the viscosity of the material; with a smaller loss modulus and smaller damping loss factor, the material approaches an ideal elastic material [34]. Figure 6a displays the results of frequency sweep experiments performed on the RBOB-FG emulsion-filled gel and FG hydrogel with varying concentrations of FG. All the gel samples with varying FG concentrations exhibited dominant elastic behavior, showing G′ > G″. The G′ value increased with increased FG concentration due to the more extensive crosslinking of the gel network. The storage moduli values (G′) of RBOB-FG emulsion-filled gel were higher than those of the FG hydrogel, suggesting that RBOBs led to stronger gels with a stiffer structure. The G′ of the samples filled with RBOBs was significantly higher than the G″, indicating that RBOBs can function as an active filler and reinforce the elastic gel structure. Figure 6b shows the G″/G′ values of the FG hydrogels and RBOB-FG emulsion-filled gels. Except for the 0.2% FG hydrogel, the G″/G′ values of which were close to 1, the rest of the samples were elastic rather than fluid. Emulsion-filled gels were structurally more stable than hydrogels, with higher G′ values, which can be attributed to the fact that the emulsion acts as an active filler in the gel to enhance gel stiffness.

According to flow measurements, the viscosity decreased as the shear rate increased from 0.1 to 100 s^−1^, reflecting progressive shear-induced breakdown of structure in RBOB-FG emulsion-filled gel (Figure 6c). Compared with FG hydrogel, the increased viscosity of RBOB-FG emulsion-filled gels became increasingly obvious with increased FG concentration. The flow curve was described by the Herschel–Bulkley model; the parameters are shown in Table 1. Both RBOB-FG emulsion-filled gels and FG hydrogel exhibited shear-thinning behavior. The apparent viscosity decreased as the shear rate increased, which can be described as shear thinning due to droplet flocculation [35]. The consistency index and flow behavior index were both influenced by the hydrocolloid concentration and molecular weights; elevated FG concentration or molecular weights increased the viscosity and therefore the consistency index of RBOB-FG emulsion-filled gel and FG hydrogel. Similar non-Newtonian mechanical behavior was previously demonstrated for other protein- and polysaccharide-stabilized emulsions [36]. As the concentration of FG increased, the yield stress of FG hydrogel increased significantly. The addition of RBOBs to FG hydrogel resulted in an RBOB-FG emulsion-filled gel with an higher yield stress compared to FG hydrogel. The fluidity of the sample decreased due to the increased viscosity. This phenomenon may have occurred because RBOB emulsion, as an active filler, increased the viscosity of the system, which is consistent with the microrheological results.

G′ and G″ were recorded during the temperature sweep process; results are shown in Figure 7. As observed, RBOB-FG emulsion-filled gels/FG hydrogels with FG concentrations of 0.2 and 0.8 wt.% presented with a sol–gel transition, exhibiting a temperature-dominated increase in both moduli and producing a crossover between G′ and G″ (gelling point) when the temperature was below 30 °C (G′ > G″) and reaching a plateau or continuing to slowly increase. The corresponding gelling temperatures (T_gel_), as well as the G′ and G″ values of the RBOB-FG emulsion-filled gel and the FG hydrogels, were determined. Compared to FG hydrogel (Figure 7a,c), in RBOB-FG emulsion-filled gels with the same FG concentration (Figure 7b,d), a gelling point delay was observed, with a higher T_gel_ value than that of the corresponding hydrogels, indicating increased collision and entanglement between the molecules in the emulsion-filled gel. Crossover between G′ and G″ (gelling point) was not observed with 2.0 wt.% FG content in RBOB-FG emulsion-filled gels (Figure 7f) and hydrogels (Figure 7e), possibly due to the high viscosity of the system. After 5 min equilibration cooling at 20 °C of the RBOB-FG emulsion-filled gel and FG hydrogel, the samples were heated up to 90 °C. Decreases in G′ and G″ of the RBOB-FG emulsion-filled gel and FG hydrogels were observed with increased temperature, except for the sample set with 2.0 wt.% FG concentration, followed by the moduli crossover G″ > G′, indicating a gel–sol transition due to the disruption of the FG-entangled chains and network. The melting temperatures (T_m_) of RBOB-FG emulsion-filled gels (Figure 7b,d) were higher than those of the hydrogel formulations with the same FG concentration (Figure 7a,c); the RBOB-FG emulsion-filled gel samples exhibited better gel properties with relatively higher recovery G′ values throughout the cooling and heating process, once again indicating that stronger networks were created in the case of the RBOB-FG emulsion-filled gel, with a gel network with higher thermal stability.

### 3.7. Tribology of RBOB-FG Emulsion-Filled Gels

The functional relationship between friction coefficient (*μ*) and sliding speed (*S*) for FG gel (0.2%, 0.8%, 2.0%) and pure water is shown in Figure 8a, and that for RBOBs and RBOB-FG emulsion-filled gel is exhibited in Figure 8b. As the results show, both pure water and FG hydrogel as hydrophilic substances exhibited typical Stribeck curves between the interface (stainless ball and the PDMS film) [37]. In the boundary lubrication regime, pure water with a *μ* plateau up to *S* = 60 mm/s and a maximum value of friction coefficient (*μ*) of between 1.42 and 1.93. Since PDMS films are hydrophobic, so water is unable to form a continuous lubricating interface. With a boundary lubrication regime up to 3.8 mm/s, *μ* values of 0.2%, 0.8%, and 2.0% FG hydrogels were 0.82–0.86, 0.42–0.47, and 0.30–0.37, respectively, suggesting that the fragments generated by the FG hydrogel were entrained into the contact surface during frictional sliding, reducing surface roughness and thus improving the lubrication.

With increased sliding speed (*S*), the *μ* value of the FG hydrogels (0.2%, 0.8%, and 2.0%) decreased to around 0.024–0.045, and those of pure water decreased to about 0.02 at a rate of 450 mm/s. In the examined sliding speed range, neither pure water nor the FG hydrogel showed a hydrodynamic lubrication regime. During the mixed lubrication regime, a reduction in the *μ* value of the FG hydrogels was observed, possibly due to the FG breaking into more fragments with increased *S* values, resulting in large amounts of FG fragments being entrained to the friction interface [38]. The emulsion-filled gel with the same FG concentration had a lower *μ* value in this regime than the FG gel due to the relatively higher G′ of the RBOB-FG emulsion-filled gel. Because the RBOB-FG emulsion-filled gel had a higher gel strength and a more stable network structure, it was not easily deformed when extruded by a perpendicular load between two contact surfaces, which increased rough surface separation and reduced friction. As shown in Figure 8b, the *μ* of the RBOBs increased from 0.04 at *S* = 150 mm/s to 0.088 at *S* = 150 mm/s during the boundary lubrication regime. According to the plate-out theory explained in [39], the emulsion droplets deformed and fractured with increasing shear force, resulting in disruption of emulsion droplets with oil spreading and the minimum *μ* value. However, no obvious oil leakage was observed after the tribology test. Because the viscoelastic RBOB droplets easily entrained, filling between the two surfaces reduced the friction at a slow sliding speed.

The “bearing balls” theory considers RBOBs as pre-emulsion droplets with a similar spherical shape and small particle size [40]. Therefore, RBOBs can act as relatively tough bearing balls and reduce friction by reducing the contact area between the tribo-pair surfaces and changing local relative motion from sliding to scrolling. The FG hydrogel matrix also reduced friction. The RBOB-FG emulsion-filled gel broke down into small pieces with increased *S* (4 < *S* < 10 mm/s), and fragments of FG matrix containing RBOBs were entrained into the contact surfaces, reducing the *μ* value. First, the leaked RBOB droplets acted as small, spherical bearings that can scroll and bear loads [41]. Then, the droplets inside the FG fragments increased the G′ of the RBOB-FG emulsion-filled gel because active filler can interact with the FG gel matrix. The lubrication mechanism for RBOB in a mixed regime and the lubrication mechanism for RBOB-FG emulsion-filled gel in a boundary regime are illustrated in Figure 8c. The lubrication mechanism of the RBOB-FG emulsion-filled gels in the boundary lubrication regime resulted in entrainment of the sample fragments and a ball-bearing effect of the oil bodies. In the regime of *S* > 90 mm/s, more fragments were entrained into the contact surfaces, and parts of the gel fragments may have squeezed and flipped over each other between the gaps, resulting in an increase in *μ*.

### 3.8. Schematic Diagram of RBOB-FG Emulsion-Filled Gels

A schematic diagram is presented to illustrate the formation of RBOB-FG emulsion-filled gels (Figure 9). Due to the dipole moment changes, the droplets of RBOBs were adsorbed around the FG molecule during the emulsion preparation, and FG transformed into an outer water phase of the RBOB emulsion [42]. RBOB droplets were filed in FG in a gel state with a network after heating and cooling, and the RBOB-FG emulsion-filled gel was formed [43]. RBOBs behaved as active filler in the matrix, promoting the integration of RBOB droplets into the gel matrix and strengthening the gel structure, as proven by its appearance, microstructural observation, and rheological properties. There was a relatively even distribution of RBOB droplets in RBOB-FG emulsion-filled gel, as shown in the CLSM micrographs. There was a low degree of aggregation in RBOB-FG emulsion-filled gel with 2 wt.% FG concentration, although not affecting the increased gel strength of the RBOB-FG emulsion-filled gel, as supported by their appearance and rheological results. This indicates that RBOB-FG emulsion-filled gel prepared with RBOBs and FG hydrogel exhibited active filling behavior as part of the network and induced a tighter network structure, as revealed by SEM observation.

## 4. Conclusions

RBOB-FG emulsion-filled gel was produced with FG as the gel matrix and filled with RBOB droplets. The result show that negatively charged FG enhanced electrostatic repulsion, and steric hindrance facilitated the physical stability of the droplets, changing the structure of RBOB protein through the formation of hydrogen bonds. The microstructure of RBOB droplets in the FG hydrogel gel matrix showed good distribution and formed an RBOB-FG emulsion-filled gel structure. RBOBs improved the storage modulus, viscoelasticity, melting, and solidification temperature, indicating that RBOBs can function as an active filler, strengthening networks. The ball-bearing effect of RBOBs and higher storage moduli values of RBOB-FG emulsion-filled gel are believed to contribute to improved lubricating properties relative to FG gels. The present work provides insights on the behaviors of RBOB emulsion-filled gel, which can be applied to design strategies for practical processes to obtain OB emulsion gels in food structuring and as fat replacements in fresh sausages, salad dressings, and bakery products.

## Figures and Tables

**Figure 1 foods-11-03110-f001:**
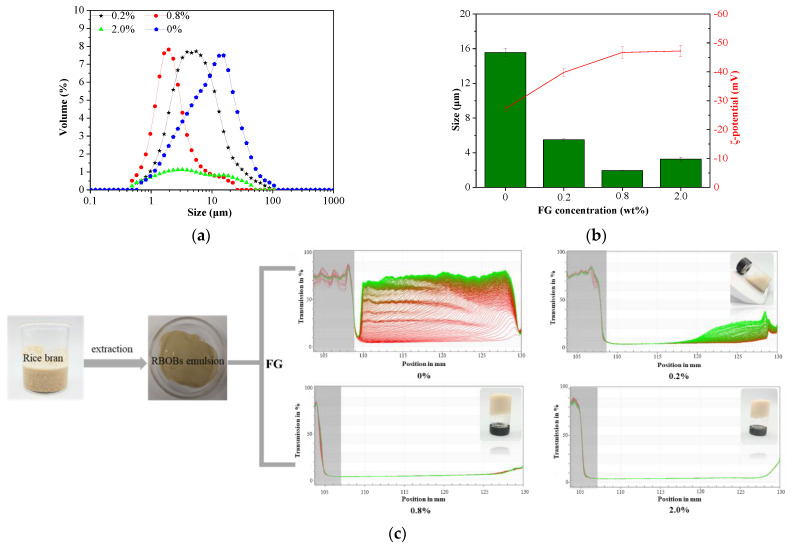
Particle size distribution (**a**), ζ-potential (**b**), and stability analysis (**c**) of the RBOB-FG emulsion-filled gels with varying FG concentrations dispersed in 5.0 wt.% RBOB emulsion.

**Figure 2 foods-11-03110-f002:**
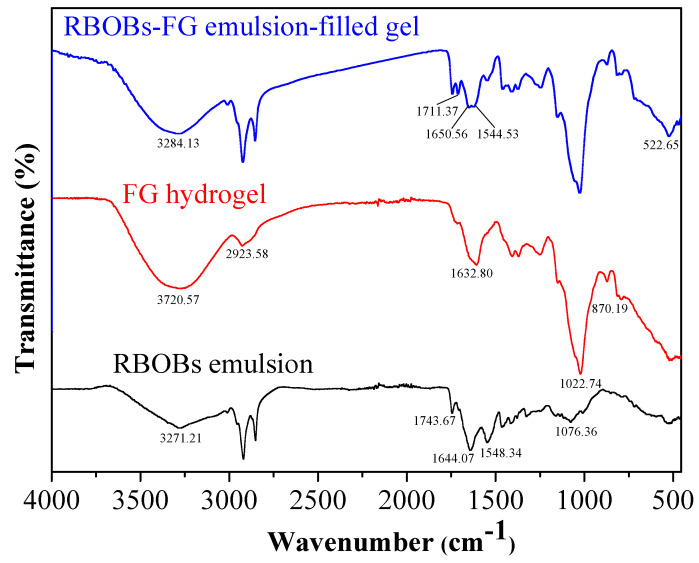
FT-IR spectra of RBOB-FG emulsion-filled gel, FG hydrogel, and RBOB emulsion.

**Figure 3 foods-11-03110-f003:**
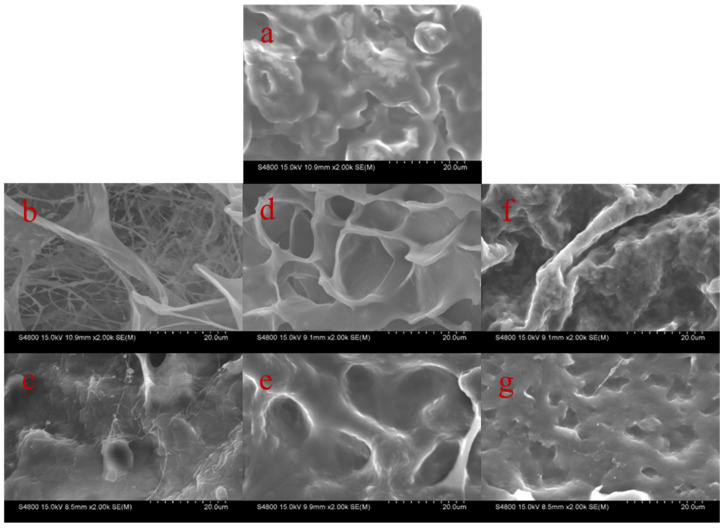
SEM of RBOB-FG emulsion-filled gel samples: RBOB emulsion (**a**), FG hydrogel (0.2 % FG) (**b**), RBOB-FG emulsion-filled gel (0.2 % FG) (**c**), FG hydrogel (0.8 % FG) (**d**), RBOB-FG emulsion-filled gel (0.8 % FG) (**e**), FG hydrogel (2.0 % FG) (**f**), and RBOB-FG emulsion-filled gel (2.0 % FG) (**g**).

**Figure 4 foods-11-03110-f004:**
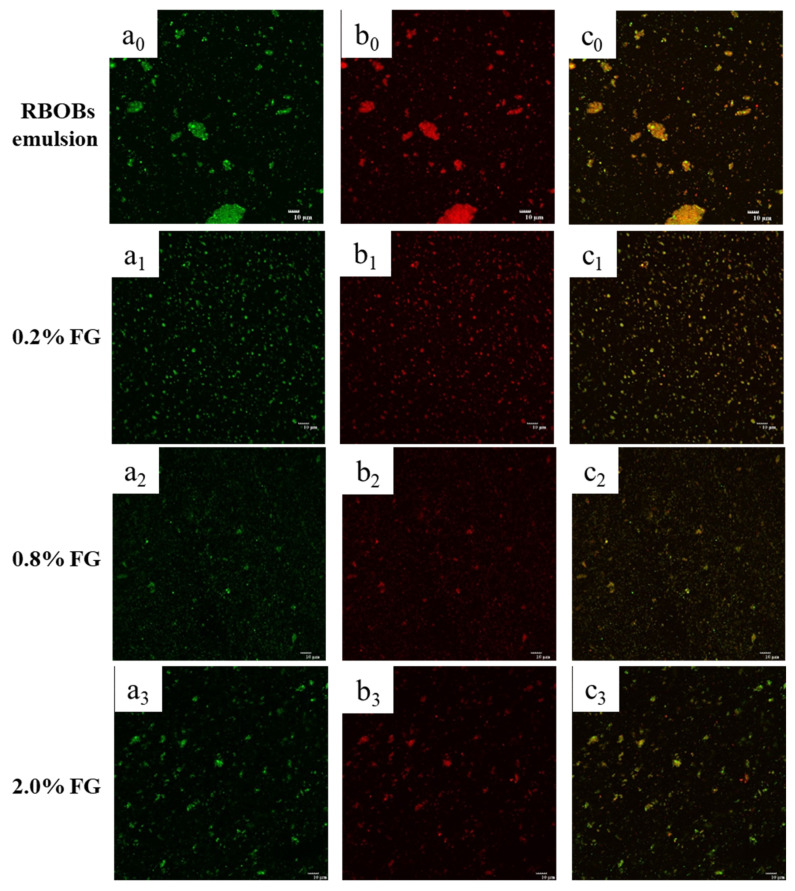
CLSM of RBOB emulsion and RBOB-FG emulsion-filled gel with varying FG concentrations: (**a_0_**–**a_3_**) are oils (stained green); (**b_0_**–**b_3_**) are proteins (stained red); and (**c_0_**–**c_3_**) are oils and proteins.

**Figure 5 foods-11-03110-f005:**
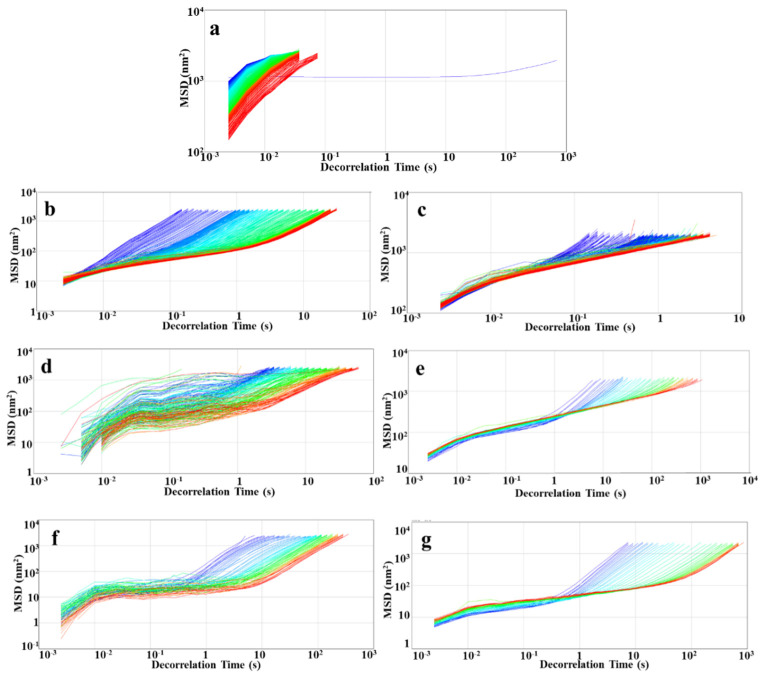
Microrheological properties of RBOB-FG emulsion-filled gel samples: mean square displacement (MSD) vs. time curves: RBOB emulsion (**a**), FG hydrogel (0.2% FG) (**b**), RBOB-FG emulsion-filled gel (0.2% FG) (**c**), FG hydrogel (0.8% FG) (**d**), RBOB-FG emulsion-filled gel (0.8% FG) (**e**), FG hydrogel (2% FG) (**f**), and RBOB-FG emulsion-filled gel (2.0% FG) (**g**).

**Figure 6 foods-11-03110-f006:**
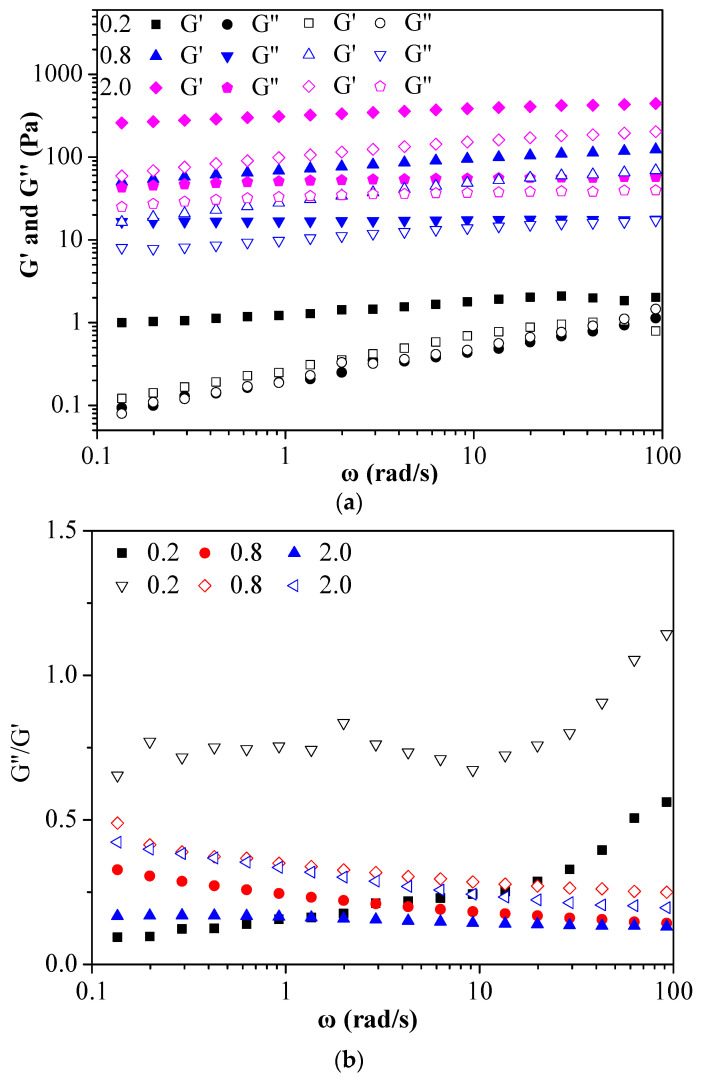
Effect of FG concentration (0.2/0.8/2.0%) on frequency sweep curves of the RBOB-FG emulsion-filled gel (solid markers), FG hydrogels (open markers) (**a**), G″/G′ values (**b**), and shear viscosity curves (**c**).

**Figure 7 foods-11-03110-f007:**
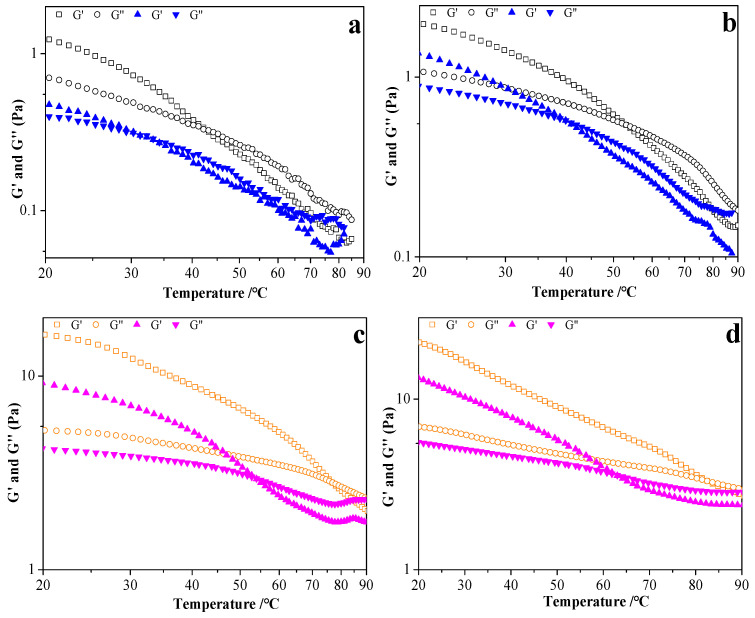
Temperature dependence of G′ and G″ moduli of RBOB-FG emulsion-filled gel during cooling (solid markers) and heating (open markers) ramps. FG hydrogel (0.2% FG) (**a**); RBOB-FG emulsion-filled gel (0.2% FG) (**b**); FG hydrogel (0.8% FG) (**c**); RBOB-FG emulsion-filled gel (0.8% FG) (**d**); FG hydrogel (2.0% FG) (**e**); RBOB-FG emulsion-filled gel (2.0% FG) (**f**).

**Figure 8 foods-11-03110-f008:**
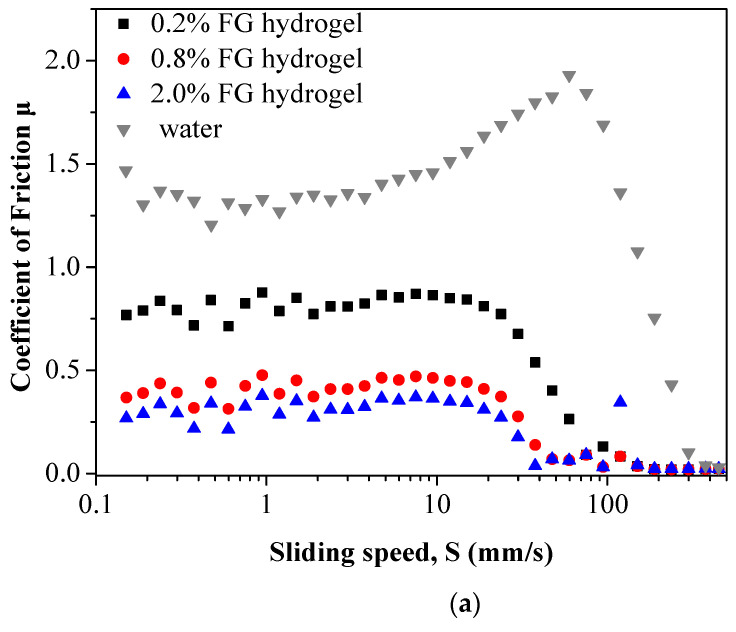
Tribology of FG gel matrix and pure water (**a**), RBOB cream and RBOB-FG emulsion-filled gels (**b**), lubrication mechanism for RBOB cream in a mixed regime, and lubrication mechanism for RBOB-FG emulsion-filled gels in a boundary regime (**c**). Droplets are represented by yellow spheres, the continuous phase is represented by blue color, balls and disks are represented in grey, and the FG gel network is represented by black curved lines.

**Figure 9 foods-11-03110-f009:**
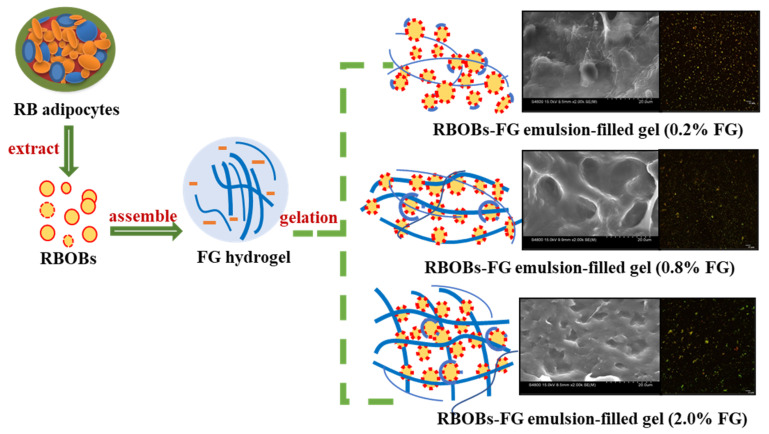
Schematic diagram of the RBOB-FG emulsion-filled gels.

**Table 1 foods-11-03110-t001:** Rheological parameters of the Herschel–Bulkley model for RBOB-FG emulsion-filled gels and FG hydrogel.

Sample	*τ* _0_	*K* (Pa S^n^)	*n*	R^2^
FG hydrogel (0.2% FG)	0.2442	0.2784	0.4365	0.9962
RBOB-FG emulsion-filled gel (0.2% FG)	0.2667	0.3066	0.3952	0.9940
FG hydrogel (0.8% FG)	11.81	4.858	0.4633	0.9521
RBOB-FG emulsion-filled gel (0.8% FG)	12.41	6.401	0.2633	0.9604
FG hydrogel (2.0% FG)	34.68	21.83	0.4804	0.9735
RBOB-FG emulsion-filled gel (2.0% FG)	78.26	21.93	0.5351	0.9581

## Data Availability

The data presented in this study are available on request from the corresponding author.

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
