# Peer review of "Stability, Structure, Rheological Properties, and Tribology of Flaxseed Gum Filled with Rice Bran Oil Bodies"

_foods, 2022, doi:10.3390/foods11193110_

Round 1

Reviewer 1 Report

- The topic is interesting and corresponds to the journal topics.

However, a major revision is required before the paper can be considered for publication as follows:

- Generally, the application of English is not acceptable in the present form, the writing style needs to be improved. A scan by a native English speaker or a Professional English Language Editing Service is necessary.

- The introduction did not include information related to other methods of extraction than aqueous extraction whit advantages and disadvantages of each method, the challenges in extraction of RBOBs, technological barriers (if they can be identified), and the market potential of the RBOB .

- Introduction: authors are asked to underline the originality of this paper (other papers that shows the potential of RBOBs for fat replacement are available).

- Experimental part: The maximum FG concentration in emulsion was 2%. Authors are asked to justify why choosing this value. The answer is maybe related to a special kind of application of flaxseed gum filled with rice bran oil bodies for food industry?

 - Figures: the quality of figure 1c should be improve (axis title and units non readable). Figure 1 a-d  mix graphs with results with preparation diagram and pictures with emulsion samples. Figure 1d (preparation diagram) is not useful.

- Conclusion: authors mention "novel emulsion gel" but I don't see the novelty of the formulations presented in this paper.

- References: double numbering for each reference.

Reviewer 2 Report

Manuscript ID: Foods_ 1896456

In the article entitled: “Stability, structure, rheological properties, and tribology of flaxseed gum filled with rice bran oil bodiesauthors examine rice bran oil bodies (RBOBs) were filled with different concentrations of flaxseed gum (FG) to construct RBOBs-FG emulsion-filled gel system. The physical properties (particle size distribution, zeta-potential, physical stability and microstructure) were measured and observed. In addition the molecular interaction of RBOBs and FG was studied (by FTIR methods). The authors also determined the rheological and the tribology properties of the RBOBs-FG emulsion-filled. This is an interesting article. The authors showed a lot of knowledge of the topic of research. They used advanced measuring techniques suitable for the adopted purpose of research.

Title

The title and the aim of the study are clearly constructed.

Abstract

The abstract includes the aim of the study, methods used in the experiment and contain the principal results and conclusions.

Introduction

The introduction describes the matter of the experiment and states the problem being investigated. Authors correctly interpreted and described the significance of the results for the research. They skillfully referred to the results of other researchers. Literature references are the most current. However, the purpose of the research should be clearly defined. Why research was undertaken and what the results obtained in them may be useful.

Methods

The data is well collected. The sampling is appropriate and adequately described.

The methods are described in detail where possible. However, in my opinion, they should be supplemented with some information:

2.3. Preparation of RBOBs-FG emulsion-filled gel and FG hydrogel

How long was the samples stored in refrigeration conditions?

2.5. Physical stability measurements

Samples spinning for 4.25 h? Immediately after preparation or after refrigerated storage? If so, after what time? What is the meaning of time interval 60 s?

2.9. Rheological properties

Were samples used to measure the apparent shear viscosity immediately after preparation or after refrigeration storage?

Were samples protected against water evaporation for temperature measurements?

Results and Discussion

Authors correctly interpreted and described the significance of the results for the research.

Conclusion

The authors correctly indicate, how the results are related to the studies.

References

Literature references are appropriate and relate to the position from the last few years.

Language

The article is correctly written. English language and style are minor spell check required.

Reviewer 3 Report

In general terms, this manuscript is scientifically sound and incorporates several analytical techniques which were properly analyzed. There are some suggestions that could make this document better:

1.   Adjust line 47: Flaxseed gum (FG) IS composed of 75% polysaccharides and 25% acid polysaccharides. Likewise, this sentence lacks a bibliographic citation
2.  Citation Cui et al (1994) (Line 49) is not reported in the references.
3.      Adjust line 51: Emulsion-filled gels are a type of emulsion gels, IN which oil droplets in the gel matrix ACT as filler particles
4.      In general, English must be revised. Several sentences have the absence of a verb which I won’t highlight here since it is the authors’ responsibility to submit a grammatically-sound manuscript. One more example (Line 54): Active filters are bounded to the gel network and contribute to gel strength whereas inactive fillers are difficult to BIND with A gel matrix, do not interact with A gel matrix, and DO not HAVE AN effect on gel viscoelasticity.
5.      In Line 149, adjust. It is Herschel, not Herschele. Revise throughout the document since this mistake is repeated several times.
6.      In Figure 3, it would be advisable if image a) was changed for one with a similar zoom compared to the others (2000 X) to have the same scale for all.
7.      In conclusion, it would be advisable to provide examples of foods in which these elements can be employed as fat replacers.

Round 2

Reviewer 1 Report

Accept in present form.